# Cytoprotective Effect of Pteryxin on Insulinoma MIN6 Cells Due to Antioxidant Enzymes Expression via Nrf2/ARE Activation

**DOI:** 10.3390/antiox12030693

**Published:** 2023-03-10

**Authors:** Junsei Taira, Ryuji Tsuda, Chika Miyagi-Shiohira, Hirofumi Noguchi, Takayuki Ogi

**Affiliations:** 1Department of Bioresources Engineering, National Institute of Technology, Okinawa College, Okinawa 905-2192, Japan; 2Department of Regenerative Medicine, Graduate School of Medicine, University of the Ryukyus, Okinawa 903-0215, Japan; 3Department of Environment and Natural Resources, Okinawa Industrial Technology Center, Okinawa 904-2234, Japan

**Keywords:** insulinoma MIN6 cells, diabetes, oxidative stress, Nrf2 activator, pteryxin, HO-1

## Abstract

The low-level antioxidant activity of pancreatic islets causes type 1 diabetes due to oxidative stress, which is also the cause of failure in the pancreatic islets’ isolation and cell transplantation. In our previous study, pteryxin was found to be a natural product as a nuclear factor-erythroid-2-related factor (Nrf2) activator. This study focused on elucidation that the potentiality of pteryxin can activate the antioxidant enzymes, even under oxidative stress, by hydrogen peroxide (H_2_O_2_). Pteryxin treated with mouse insulinoma MIN6 cells was enhanced the antioxidant gene expressions in the ARE (antioxidant response element) region for HO-1 (Heme Oxygenase-1), GCLC (Glutamate-cysteine ligase catalytic subunit), SOD1 (Super Oxide dismutase1), and Trxr1 (Thioredoxin reductase1), and those enzymes were also expressed during the nuclei transference of cytoplasmic Nrf2. In fact, the cells exposed to H_2_O_2_ concentrations of a half-cell lethal in the presence of pteryxin were then induced main antioxidant enzymes, HO-1, GCLC, and Trxr1 in the ARE region. The increased glutathione (GSH) levels associated with the GCLC expression also suggested to be cytoprotective against oxidative stress by activating the redox-metabolizing enzymes involving their increased antioxidant activity in the cells. In addition, Akt is a modulator for Nrf2, which may be responsible for the Nrf2 activation. These results allowed us to consider whether pteryxin or its synthesized congeners, an Nrf2 activator, is a potential preservative agent against islet isolation during cell transplantation.

## 1. Introduction

Oxidative stress, with the excess production of reactive oxygen species (ROS), is related to the increased risk of developing several diseases, including obesity and diabetes mellitus [1]. Oxidative stress generates in the diabetic status and may contribute to the progressive pancreatic β-cell dysfunction in diabetes due to the low expression levels of the antioxidant enzymes [2]. Thus, the pancreatic β-cells that express low levels of many antioxidant enzymes, such as catalase, superoxide dismutase, and glutathione peroxidase, are hypothesized to be susceptible to oxidative damage induced by ROS associated with diabetes mellitus [2,3]. Oxidative stress is also a major cause of islet damage and loss during the islet isolation process or preservation of isolated cells. This is particularly problematic in the pancreatic cell transplantation, as it significantly affects the survival of β-cells [4]. Therefore, the preservation solution for the cell transplantation has been required in the medical practice field [5]. 

The antioxidant response element (ARE) is activated by the nuclear factor E2-related factor2 (Nrf2) in a major cellular defense mechanism against the oxidative stress or response to electrophilic chemicals. The Nrf2 dissociates from the Kelch-like ECH-associated protein 1 (Keap1) by electrophiles and oxidative stress [6]. The Nrf2 performs a significant role in the regulation of adipocyte differentiation, obesity, and insulin resistance [7]. In addition, the Nrf2 induction restored insulin secretion from pancreatic β-cells due to suppression of the accumulation of intracellular ROS in the isolated islets and pancreatic B-cells through alteration of the gene expression related to the antioxidant, energy consumption, and gluconeogenesis in metabolic tissues [8]. Another Nrf2 study has revealed that the Keap1-Nrf2 system is a key regulator in the protection of pancreatic β-cells as it preserves the islet size by both enhancement of the β-cell proliferation and repression of β-cell apoptosis in diabetic model mice under oxidative or nitrative stress [9]. Therefore, a more effective Nrf2 activator is required to reduce the oxidative stress, particularly the low-level of the antioxidant status of the pancreatic islets. There is a significant potential for clinical applications of the Nrf2 activators in patients with diabetes. Based on these medicinal backgrounds, the Nrf2 activators, such as CDDO-9,11-dihydro-trifluoroethyl amide (Dh404), and dimethyl fumarate (DMF), have been significantly associated with inflammation involving oxidative stress and diseases of the heart, kidney, and pancreas [10,11,12]. They increased the expression of the key antioxidant enzymes, decreased inflammatory mediators in the islets, and conferred protection against oxidative stress in β-cells. 

Recent studies have demonstrated that the ethanol (EtOH) extract of *P. japonicum* has an anti-obesity effect and it contains coumarin-related compounds, including pteryxin, that affect diabetes and obesity, both of which are bioaccessible to the systemic tissues [13,14,15,16,17,18]. For example, anti-diabetic and anti-obesity effects of cis-3’,4’-diisovalerylkhellactone suppressed adipocyte differentiation and stimulated glucose uptake via activation of AMPK and down-regulation of adipogenic transcription factors [17]. Pteryxin inhibited the transcription factors for lipid synthesis in the differentiated adipocytes and in hepatocytes [18]. Additionally, our recent study found the highest Nrf2 activity in the EtOH extract of *P. Japonicum* Thunb leaves, and its Nrf2 active compound was identical to that of pteryxin, then it induced the expression of the antioxidant protein, HO-1 [19]. In addition, the Nrf2 active function, due to pteryxin, was suggested to hold electrophillicity due to the α,β-carbonyl and/or substituted acyl groups in the molecule modulating the dissociation of Nrf2 from the Keap 1. The highest Nrf2-activating compound, pteryxin, will be expected as a preservation solution for transplantation under oxidative stress state. Therefore, this study placed aim to elucidate the antioxidant ability of pteryxin inducing the expressions of antioxidant gene and enzyme on the ARE region in insulinoma MIN6 cells, and its potentiality as a transplant preservation solution to reduce the cell damage caused by the oxidative stress in cell transplantation.

## 2. Materials and Methods

### 2.1. Materials

Primers for HO-1 (Heme Oxygenase-1), Nqo1 (NAD(P)H dehydrogenase quinone 1), Akt (Protein Kinase B), GCLC (Glutamate-cysteine ligase catalytic subunit), GST (Glutathione *S*-transferase), SOD1 (Super Oxide Dismutase-1), Snxin1 (Sorting nexin-1), Trxr1 (Thioredoxin reductase1), Bcl-xL (B-cell lymphoma-extra-large), and GAPDH (Glyceraldehyde-3-phosphate dehydrogenase) are commercially available (Assays-on-Demand Gene Expression Products, Thermo Fisher Scientific, Waltham, MA, USA). Antibodies of Nrf2, HO-1, Akt, and GAPDH were obtained from Cell Signaling Technology (Danvers, MA, USA), and GCLC and Trxr1 were purchased form Cosmo Bio Co., Ltd. (Tokyo, Japan). SOD1 and Goat anti-rabbit IgG H&L were obtained from Abcam Plc. (Cambridge, UK). 

### 2.2. Preparation of Pteryxin

In this study, pterixin was isolated using a supercritical fluid extraction method to obtain pterixin in a convenient manner. Pteryxin was isolated from dried leaf powder of *P. japonicum* [19]. The dried leaf powder (150 g) of *P. japonicum* was extracted with supercritical carbon dioxide using the Supercritical Fluid Extraction Screening System (X-10-05. Thermo Separation Products, West Palm Beach, MA, USA) at 30 MPa and 43 °C for 3 h, then the extract of 133 mg was obtained. The extract was separated using a centrifugal chromatography with a two-phase solvent system of n-hexane/chloroform/70% methanol (9:1:10 in *v/v/v*) (Easy-PREPccc, coil column 318 mL, Kutuwa Sangyo, Hiroshima, Japan), The lower layer (mobile phase) was eluted at 3.0 mL/min at 1110 rpm, then the crude pteryxin extract (41.6 mg) was obtained (retention time at 75–90 min,). The crude pteryxin fraction was purified by a reversed-phase chromatography column (XBridge C18 column, 150 × 19 mm, I.D., 5 μm particle size, Waters Corp., Milford, MA, USA) by formic acid/H_2_O/acetonitrile (0.1/55/45 in *v/v/v*) elution at a flow rate of 12.0 mL/min using a HPLC apparatus (PU-980 HPLC pump, Japan Spectroscopic Corporation, Tokyo, Japan). Finally, 32.8 mg of pteryxin was obtained as a purified product. 

### 2.3. Analysis of Pteryxin

The purity of the isolated pteryxin was determined by the isocratic condition with a mobile phase consisting of acetonitrile/H_2_O (45:55) at a flow rate of 0.4 mL/min for 5 min by a quadrupole LC/MS/MS (Xevo TQD equipped with H-class and eλ PDA detector, Waters Corp., Milford, MA, USA) on a reversed phase chromatography column (ACQUITY UPLC BEH C18, 50 × 2.1 mm I.D., 1.7 µM particle size, Waters Corp.) at 40 °C.

The structure of pteryxin was confirmed as previous procedures using ^1^H and ^13^C-NMR spectra (Avance III HD Ascend 400 MHz spectrometer, Bruker Billerica, MA, USA) [13].

### 2.4. Cell Culture 

Mouse insulinoma MIN6 cells were cultured in DMEM medium (including 10% FBS, 100 U/mL penicillin, and 100 µg/mL streptomycin) at 37 °C in a 5% CO_2_ atmosphere.

### 2.5. Immunohistochemistry

Cells were treated with pterixin (10 μM and 50 μM, respectively) for 1 h, then the Nrf2 translocation from the cytoplasm to the nucleus due to pterixin in the cells was examined as follows. Briefly, the cells were fixed with 4% paraformaldehyde in PBS, then they were blocked using 20% AquaBlock (EastCoast Bio, MO, USA) for 30 min at room temperature. After washing the cells with PBS and incubated at 4 °C with the anti-Nrf2 antibody (1:100), they were subsequently incubated with Goat anti-rabbit IgG H&L (1:200) for 1 h. at room temperature. The cells were treated with mounting medium to detect the fluorescence emitted by the DAPI (Vector Laboratories, Peterborough, UK).

### 2.6. Quantitative Polymerase Chain Reaction (qPCR)/Reverse Transcription PCR (RT-PCR)

Cells (1.0 × 10^5^ cells/mL) were pre-cultured overnight. The pre-cultured cells with or without pterixin (2 and 50 μM) were incubated for 24 h, then the total RNA was extracted from cells with or without test compounds by a RNeasy Mini Kit (QIAGEN, Venlo, Netherlands) using commercially available procedures. Briefly, the RNA extract (2.5 μg) was heated for three minutes at 85 °C, then it was reverse-transcribed into cDNA using Superscript II RNase H-RT (Invitrogen, Waltham, MA, USA). Polymerization of 20 ng cDNA was manually carried out using DNA polymerase (Invitrogen) under amplification cycles for denaturation at 94 °C for 1 min, annealing at 57–62 °C for 1 min, and extension at 72 °C for 1 min with a final extension step at 72 °C for 10 min (Perkin-Elmer 9700 Thermocycler, Perkin Elmer, Inc., Waltham, MA, USA). Quantification of the *m*RNA levels was carried using a TaqMan real-time PCR system, according to the manufacturer’s instructions (Applied Biosystems, Inc., Waltham, MA, USA). The PCR was performed for 40 cycles, including the initial step at 50 °C for 2 min and at 95 °C for 10 min for denaturation, 15 s at 95 °C, annealing/extension, and 1 min at 60 °C. Each *m*RNA expression was normalized by the GAPDH *m*RNA expression level. 

### 2.7. Cytotoxicity

The cell viability treatment, with or without a test sample in a well, was examined by an MTT assay, as previously reported [14]. Cells (1.0 × 10^5^ cells/mL) were pre-cultured overnight. Then, the cells were treated with or without pteryxin (1 μM, 5 μM, and 10 μM, respectively) for 24 h. After the culture, MTT (0.05%) was added to each well and incubated for 3 h. The formazan reduced from the MTT was extracted with DMSO (100 µL) and was determined as an index of the surviving cells at 570 nm using a microplate reader (BIO-RAD Model 550, BIO-RAD, Hercules, CA, USA).

### 2.8. Glutathione (GSH) Content

Cells (1.0 × 10^5^ cells/mL) were pre-cultured overnight. The pre-cultured cells with or without a test sample was incubated for 24 h. The total glutathione including GSH and GSSG in the cell lysate was determined by available instructions using the GSH/GSSG-Glo™ Assay kit (Promega K. K., Madison, WI, USA) and a microplate reader (GLOMAX MULTI Detection system, Promega K. K., Madison, WI, USA). Briefly, the determination of the total glutathione measured the luciferin production from the GSH-dependent GSH probe was reduced by glutathione-*S*-transferase with the firefly luciferase reaction. In addition, GSSG was measured by adding GSH blocking reagent for the lysate, separating only the oxidized GSSG, then the GSSG content was quantified by a luminescence reaction with luciferase. The luminescence intensity depending on the amount of the total GSH and GSSG was measured using a microplate reader. The standard curve of GSH (0–16 µM) was made by luminescence for each GSH and GSSG concentration, then the GSH content of test sample concentrations were expressed as the amount of the total GSH except for the GSSG content. 

### 2.9. Protein Expression with or without H_2_O_2_ Treatment

The expressions of antioxidant proteins due to pteryxin, with or without H_2_O_2_ (100 μM) treatment in a well, was examined by Western blot analysis. Cells (1.0 × 10^5^ cells/mL) were pre-cultured overnight. The pre-cultured cells were incubated with pteryxin (1 μM, 5 μM, and 10 μM, respectively) or without for 24 h, then cells were exposed to H_2_O_2_ (100 μM) for 24 h. The cells were washed with PBS, then treated with the lysis buffer. The cellular lysates were centrifuged at 13,800 g for 5 min. The total cellular extracts were separated on SDS-polyacrylamide gels (4–12% SDS-polyacrylamide, Invitrogen) and transferred to a nitrocellulose membrane (iBlot Gel Transfer Mini, Invitrogen) using an iBlot Gel Transfer Device (Invitrogen). The protein detection was carried out using an immunodetection system (Invitrogen) with the antibodies.

### 2.10. Statistical Analysis

The data were expressed as the means ± SD. F test was used the degree of variability in two groups, then they were compared using the Student’s *t*-test. The differences between each group were considered to be significant, **p* < 0.05 and ***p* < 0.01.

## 3. Results

### 3.1. Pteryxin and Its Cell Viability

Pteryxin was isolated from the leaf extract of *P. Japonicum* Thunb using supercritical carbon dioxide by super critical extraction method (Figure 1a). Pteryxin is an angular type khellacton coumarin with substituted acyl groups [19]. The cytotoxicity of pteryxin (2 μM, 10 μM, and 50 μM, respectively) was evaluated for 24 h incubation in MIN6 cells. No cytotoxicity of pteryxin was observed in the range of the test concentrations (Figure 1b). The similar result was obtained from mouse macrophages RAW264.7 cells [19]. Thus, these pteryxin concentrations as a reference were used throughout this study. 

### 3.2. Nrf2 Translocation from Cytoplasm to Nucleus by Pteryxin

The Nrf2 translocation from the cytoplasm to the nucleus due to pteryxin in the cells was examined. Cells treated and untreated with pteryxin were stained with the Nrf2 antibody (green color) and DAPI (blue color) as shown in Figure 2. The Nrf2 translocation from the cytoplasm to the nucleus was clearly observed in the confocal microscope with the pteryxin (10 μM and 50 μM) in the 1 hr treated cells. The previous study supported the result that the Nrf2-ARE signaling using the reporter assay was activated in the presence of pteryxin [13]. 

### 3.3. Antioxidant Genes Expression

The gene expression in the presence of pteryxin was investigated in the ARE regions and the genes of HO-1, GCLC, SOD1, Srxn1, Trxr1, Nqo1, and Gstp1 were detected (Figure 3). The expression of HO-1 was remarkably higher than those of the other genes and the Keap1 expression was included. This result suggested that the pteryxin as the Nrf2 activator strongly acted on Keap1, then the Nrf2 signaling activated GCLC, SOD1 and Trxr1 in the ARE regions. In addition, pteryxin also expressed the antiapoptosis gene, Bcl-xL and its upper regulating protein, Akt, contributing to cell survival.

### 3.4. Antioxidant Enzymes Expression

The antioxidant protein expression in the presence of pteryxin was investigated concerning the antioxidant enzymes in the ARE regions: i.e., HO-1, GCLC, SOD1, and Trxr1, including Keap1 and Nrf2 (Figure 4). As shown in Figure 4, the antioxidant enzymes expressed in the presence of pteryxin in a dose dependent-treated concentration. A weak the Nrf2 expression was also detected, then the Nrf2 dissociation proceeded due to the pteryxin modulation, resulting in the decreased Keap1 expression. These enzyme expressions are almost related to those of the antioxidant gene expressions in the ARE region (Figure 3).

### 3.5. GSH Content

The tripeptide GSH participates in many critical cellular functions, including antioxidant defense and cell growth. GCLC is a key catalytic enzyme that produced GSH from glutamate cysteine and glycine. The GSH content was determined with or without pteryxin. As shown in Figure 5, the GSH content was enhanced in the presence of pteryxin, which is related to that of the GCLC expression (Figure 4d). This result indicated that the production of GSH contributes to the cytoprotective effect due to the cellular redox reactions, such as its antioxidant activity for the reactive oxygen and nitrogen species and thioether formation [20,21,22].

### 3.6. Antioxidant Enzyme Expressions under H_2_O_2-_Treated Cells

It was found that pteryxin has the activating effects of the antioxidant enzyme (Figure 3). Based on this fact, the effect of pteryxin was evaluated in cells under oxidative stress caused by H_2_O_2_ treatment. The results showed that the expressions GCLC, Trxr1, and HO-1 antioxidant enzymes were observed in the presence of pteryxin (Figure 6). Thus, the antioxidant enzyme expression activity of pteryxin was shown to be useful against oxidative stress.

## 4. Discussion

The low-level antioxidant activity of pancreatic islets causes type 1 diabetes due to oxidative stress, thus the cause of failure is the pancreatic islets isolation and cell transplantation. This is particularly problematic in pancreatic cell transplantation as it significantly affects the survival of the β-cells [2,3]. Therefore, the preservation solution for the cell transportation has been required in the medical practice field. The ARE region is activated by the Nrf2 in a major cellular defense mechanism against oxidative stress. 

Previous studies have reported the Nrf2 activators of dh404 and methyl fumarate, which prevent damage during pancreatic oxidative stress via the Nrf2 pathway. These Nrf2 activators markedly increase the expression of the key major antioxidant enzymes and protect β-cells by reducing inflammatory mediators [23,24]. In this study, pteryxin as a natural a Nrf2 activator was used mouse insulinoma MIN6 cells (Figure 1) [19]. As shown in Figure 2, the dissociated Nrf2 due to pteryxin transferred from the cytoplasm to the nucleus and promoted the expression of the genes, such as HO-1, GCLC, SOD1, Srxn1, and Trxr1 encoded in the ARE region (Figure 3). Whereas pteryxin had no effect on the gene expressions of Gstp1 and Nqo1, which perform an important role in detoxification by catalyzing the conjugation of quinones and electrophilic compounds. Pteryxin is an angular-type khellacton coumarin, which has an electrophilicity due to the α,β-carbonyl and/or substituted acyl groups in the molecule modulating the dissociation of Nrf2 from the Keap1, but pteryxin was not detoxified; therefore, it could contribute to the Nrf2 activation in the cells. In addition, the expression of HO-1 involved in the Keap1 expression was remarkably higher than those of the other gene expressions which suggested that pteryxin acts to dissociate the Nrf2 from the Keap1. Consequently, pteryxin would activate the Nrf2-ARE signaling in our previous report [19]. 

When the cysteine residue in the Keap1 is oxidized by an electrophile, the Nrf2 part from Keap1 binds to the ARE region in the DNA sequences. A multitude of Nrf2 inducers have been reported, most of which are electrophilic and directly react with the cysteine thiol groups in Keap1 [25]. The functional significance of these cysteine residues, Cys151, Cys273, and Cys288, perform a fundamental role in the sensing of the electrophilic Nrf2 activators [26]. Based on the functional necessity of these three cysteine residues, various Nrf2 activators, such as sulforaphane, dimethyl fumarate, and 1-[2-cyano-3,12-dioxooleana-1,9(11)-dien-28-oyl] imidazole, are Cys151-dependent compounds and 15-deoxy-D12,14-prostaglandin J2 (15d-PGJ2) combines with Cys288 also consisting of 4-hydroxy-nonenal, sodium meta-arsenite, and 9-nitro-octadec-9-enoic acid, can react with any of the three sensor cysteines of Cys151, Cys273, and Cys288 [27,28]. Pteryxin has a similar structure to dimethyl fumarate, which has the α,β-carbonyl moiety its molecule, therefore target of pteryxin may be Cys151 on the sequences of Keap1. 

Pteryxin was expressed the antioxidant enzymes related to the following genes: HO-1, GCLC, and Trxr1 involving Akt and the antioxidant gene expressions (Figure 5). GSH was enhanced in the presence of pteryxin, which is related the expression of the enzyme, GCLC (Figure 4d). GSH is an important intracellular peptide with multiple functions ranging from antioxidant defense to the modulation of cell proliferation. The production of GSH contributes to the cytoprotective effect due to cellular redox reactions against the excess production of ROS and nitrogen oxide, and is also an SH donor for various enzymes [20]. GSH is synthesized in the cytosol of all mammalian cells in a tightly regulated manner. The major determinants of the GSH synthesis are the availability of cysteine, the sulfur amino acid precursor, and the activity of the rate-limiting enzyme, GCLC. Many conditions alter the GSH level via changes in the GCLC activity and GCLC gene expression (Figure 3 and Figure 4) [21,22]. These include oxidative stress, activators of the Phase II detoxifying enzymes, antioxidants, drug-resistant tumor cell lines, hormones, cell proliferation, and diabetes mellitus. 

Based on these results, the effect of pteryxin was examined in cells under oxidative stress caused by the H_2_O_2_ treatment, then the expression of GCLC, Trxr1, and HO-1 was enhanced. A previous study showed an enhanced HO-1 expression in dh404, the Nrf2 activator treated islets, but not in the other main antioxidants, suggesting that the antioxidant potentiality of pteryxin may be a high activity (Figure 6). Therefore, the potential cytoprotective activity by upregulating the genes mechanism against cell damage causing oxidative stress. In other words, pteryxin was shown to be effective in the cytoprotective action for pancreatic cells during islet cell transplantation. The potential of pteryxin or similar chemical synthesis as a protective agent in the transplantation may be a candidate dependent on the research progress in the future.

## 5. Conclusions

This study showed that the Nrf2 activator, pteryxin, has the potential to prohibit cellular damage related to the expression of antioxidant genes and enzymes on the ARE region in the nuclei of insulinoma MIN6 cells. In addition, the effect of pteryxin was examined in cells under oxidative stress caused by the H_2_O_2_ treatment, then the expression of enzymes, such as GCLC, Trxr1, and HO-1 was enhanced. This suggests that pteryxin or its synthesized congeners may be useful as a preservation reagent during islet cell transplantation.

## Figures and Tables

**Figure 1 antioxidants-12-00693-f001:**
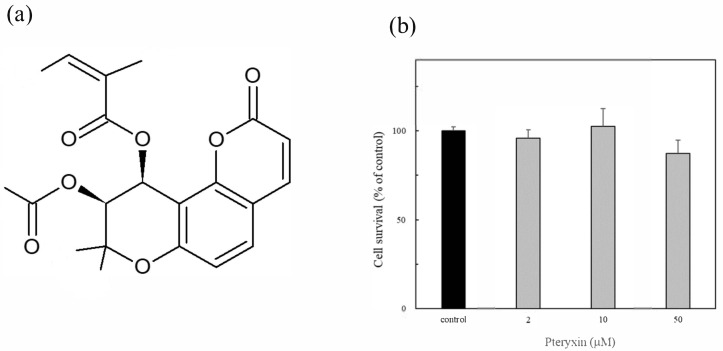
Pteryxin and its cytotoxicity in insulinoma MIN6 cells. (**a**) Chemical structure of pteryxin. (**b**) Cell viability treated with at various pteryxin concentrations (2 μM, 10 μM, and 50 μM, respectively).

**Figure 2 antioxidants-12-00693-f002:**
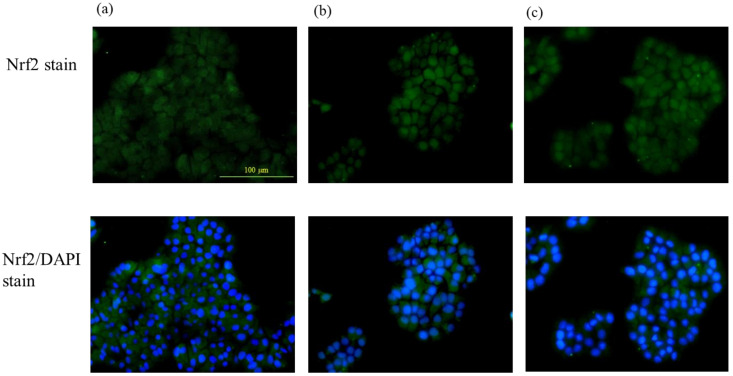
The Nrf2 translocation to the nucleus from the cytoplasm in the presence of pteryxin in insulinoma MIN6 cells. Cells treated and untreated with pteryxin were stained with Nrf2 antibody (green color) and DAPI (blue color). (**a**) Control cells (DMSO) without pteryxin, (**b**) and (**c**) treated cells with pteryxin for 10 μM and 50 μM, respectively.

**Figure 3 antioxidants-12-00693-f003:**
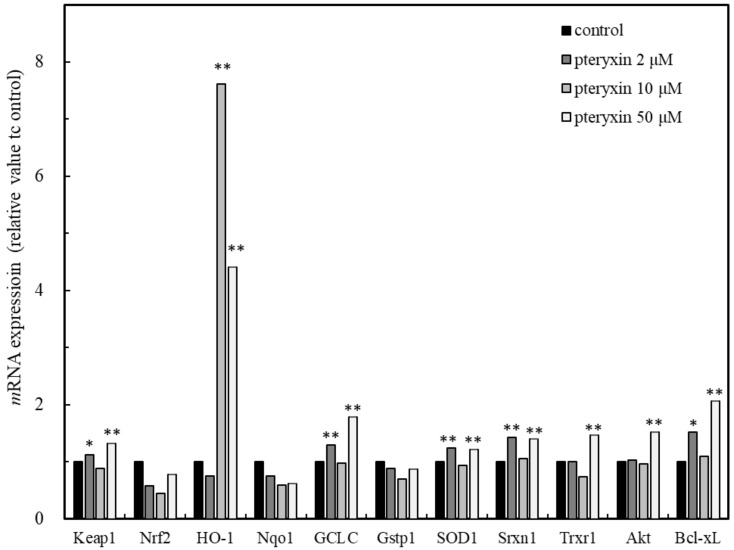
Main genes expression in ARE region including Akt and Bcl-xL due to pteryxin in insulinoma MIN6 cells. Genes expression on ARE region, such as HO-1, Gclc, SOD 1, Srxn1, Txrx1, Nqo1, and Gst1 involving apoptosis regulation related genes of Akt and Bcl-xL in the presence of pteryxin. The expression level of *m*RNA was shown as % of individual control. Data were expressed as mean ± SD, and the significant difference was analyzed by the student’s *t*-test. * *p* < 0.05 and ** *p* < 0.01 indicated as a significant difference from control.

**Figure 4 antioxidants-12-00693-f004:**
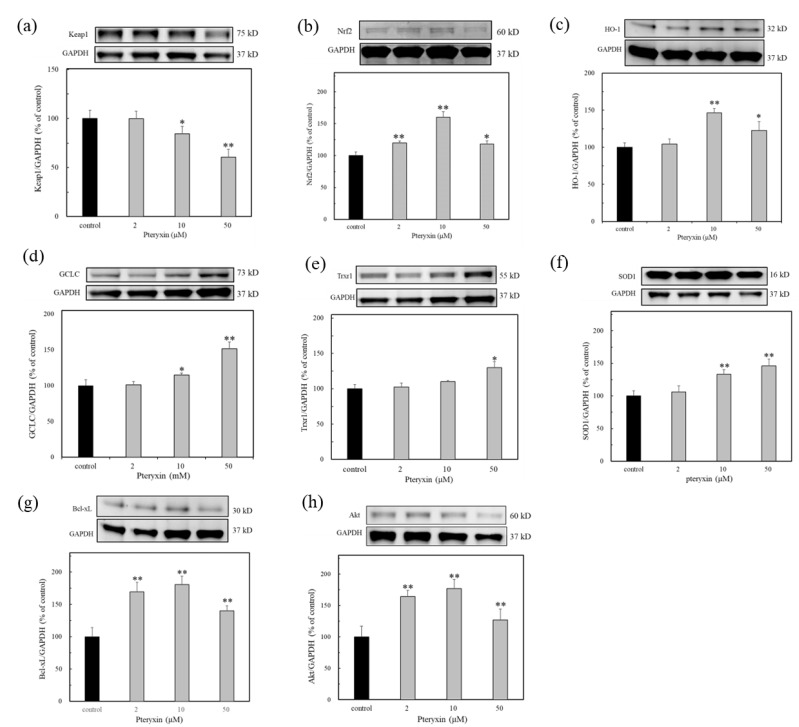
The antioxidant enzyme expression in ARE region due to pteryxin treatment for 24 h. In insulinoma MIN6 cells. (**a**) Keap1, (**b**) Nrf2, (**c**) HO-1, (**d**) GCLC, (**e**) Trxr1, (**f**) SOD1, (**g**) Bcl-xL, and (**h**) Akt. Data were expressed as mean ± SD, and the significant difference was analyzed by the student’s *t*-test. * *p* < 0.05 and ** *p* < 0.01 indicated as a significant difference from control.

**Figure 5 antioxidants-12-00693-f005:**
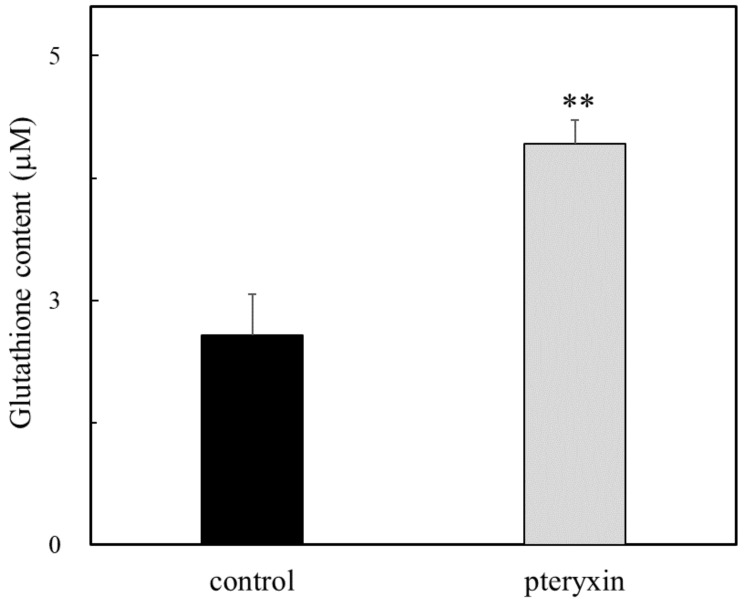
The GSH content in the presence of pteryxin. The content of GSH was determined with or without pteryxin (50 μM) for 24 h incubation as shown in the text. Data were expressed as mean ± SD, and the significant difference was analyzed by the student’s *t*-test. ** *p* < 0.01 indicated as a significant difference from control.

**Figure 6 antioxidants-12-00693-f006:**
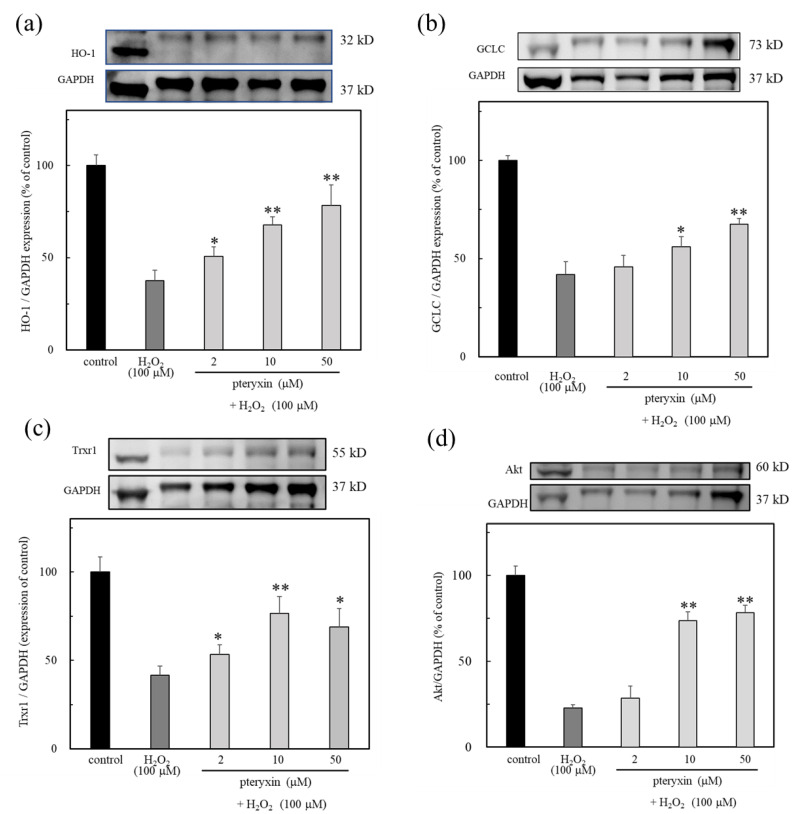
The antioxidant enzymes expression due to pteryxin under H_2_O_2_ treated cells for 24 hrs. The results showed that the expression of HO-1, GCLC, Trxr1, and Akt were expressed in the presence of pteryxin. (**a**) HO-1, (**b**) GCLC, (**c**) Trxr1, and (**d**) Akt. Data were expressed as mean ± SD, and the significant difference was analyzed by the student’s *t*-test. * *p* < 0.05 and ** *p* < 0.01 indicated as a significant difference from control H_2_O_2_ treated cells.

## Data Availability

The data presented in this study are available in article.

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
