# Peer review of "Cytoprotective Effect of Pteryxin on Insulinoma MIN6 Cells Due to Antioxidant Enzymes Expression via Nrf2/ARE Activation"

_antioxidants, 2023, doi:10.3390/antiox12030693_

Round 1
Reviewer 1 Report
The authors investigated the protective effect of pterycin, a plant derivative for transplantation. While the authors built on previous findings, this submission has several concerns.
1) A thorough read-through is necessary. For example, line 53 the period is before the [8]. Line 239 small numbers 2.
2) Be consistent with your wording. While b cells were used in the introduction, in line 53 they used B cells
3) Under Materials & Methods, describe where or how pterycin was obtained. Cited your previous publication if this is the case.
4) Why the use of different concentrations of pteryxin. The authors sometimes use 10 and 50 uM and sometimes 2 and 50 uM. Is there a specific reason?
5) The main concern is the lack of experimental details. The immunohistochemistry to determine the effects of pterixin on the Nrf2 translocation was incubated for 1 hour. In the MTT cytotoxicity assay the incubation was for 24 hours. Nothing is described for the GSH content experiment, the antioxidant gene expression, and the data for Figure 4.
6) Similarly, Figures 5 and 6 lack details about incubation times.
Author Response
Response to reviewer 1:
We will appreciate for taking the time to provide us with your kind comments. We have responded to your comments as follows.
#1 Comments and Suggestions for Authors
The authors investigated the protective effect of pterycin, a plant derivative for transplantation. While the authors built on previous findings, this submission has several concerns.
- A thorough read-through is necessary. For example, line 53 the period is before the [8]. Line 239 small numbers 2.
Response: As you suggested revised on lines of 53 and 239. Also, we checked through the manuscript.
- Be consistent with your wording. While b cells were used in the introduction, in line 53 they used B cells.
Response: B cell changed b cell in line 54.
- Under Materials & Methods, describe where or how pterycin was obtained. Cited your previous publication if this is the case.
Response: This time, pterixin was isolated using a supercritical fluid extraction method to obtain pterixin in a convenient manner. The procedures added in the text (lines 99-123).
- Why the use of different concentrations of pteryxin. The authors sometimes use 10 and 50 mM and sometimes 2 and 50 m Is there a specific reason?
Response:
In Fig. 2, when the pteryxin was treated with 10 and 50 mM concentrations without toxicity, the nuclear translocation and the difference dependent concentrations was clearly shown.
In Fig. 3, the gene expressions of pteryxin were clearly detected at 50 mM treatment under comparable level of each gene. This time, the graph was remade with adding 10 mM data.
- The main concern is the lack of experimental details. The immunohistochemistry to determine the effects of pterixin on the Nrf2 translocation was incubated for 1 hour. In the MTT cytotoxicity assay the incubation was for 24 hours. Nothing is described for the GSH content experiment, the antioxidant gene expression, and the data for Figure 4.
Response: As you suggested, incubation time for the GSH content experiment, the antioxidant gene expression and western Blot analysis for Figure 4 added in the text and Figures legend (letters in red).
- Similarly, Figures 5 and 6 lack details about incubation times
Response: As you suggested, the incubation time added in the legends of Figures 5, 6 (letters in red).
Reviewer 2 Report
The manuscript entitled "Cytoprotective effect of pteryxin on insulinoma MIN6 cells due to antioxidant enzymes expression via Nrf2/ARE activation" bring new therapeutic methods in order to help for islet cells transplantation which can be vital for diabetes mellitus patients. Therefore is important to have protective mechanism for islet cells viability preservation.
The following observation have to be made:
Introduction
Please offer more information about pteryxin, the most important therapeutic molecule in your study. The plant composition description would be also useful, and the most important please emphasize if there are already any other studies about this plant and pterixin molecule properties. The Introduction chapter has to contain the "state of the art" of your research. I would suggest to rewrite the Introduction chapter in order to contain more details about this plant and pteryxin curative properties.
The aim of the study has to be written clear at the end of Introduction as the objective of the study, not as the results you already know. Please reformulate this part of Introduction
Methods
Please offer the Ethic Committee Approval in the "Material and Methods" chapter.
Regarding the statistical methods you apply, were all the data with normal distribution to use only Student's test? If yes, you must specify this.
Conclusions
Please write the conclusions with more details. Based on your results you need to mention the molecules influenced by pteryxin action and how. Anyhow the first sentence of your conclusions (line 307-308) is confusing, please reformulate it.
Author Response
Response to reviewer 2:
We will appreciate for taking the time to provide us with your kind comments. We have responded to your comments as follows.
#2 Comments and Suggestions for Authors
The manuscript entitled "Cytoprotective effect of pteryxin on insulinoma MIN6 cells due to antioxidant enzymes expression via Nrf2/ARE activation" bring new therapeutic methods in order to help for islet cells transplantation which can be vital for diabetes mellitus patients. Therefore is important to have protective mechanism for islet cells viability preservation.
The following observation have to be made:
Introduction
Please offer more information about pteryxin, the most important therapeutic molecule in your study. The plant composition description would be also useful, and the most important please emphasize if there are already any other studies about this plant and pterixin molecule properties. The Introduction chapter has to contain the "state of the art" of your research. I would suggest to rewrite the Introduction chapter in order to contain more details about this plant and pteryxin curative properties.
The aim of the study has to be written clear at the end of Introduction as the objective of the study, not as the results you already know. Please reformulate this part of Introduction
Response:
As you suggested, the introduction added sentences include the detail information of plant, function of pteryxin through the "state of the art" (lines in red). In addition, the aim of the study clearly expressed at the end of Introduction (lines 80-84 in red)
Methods
Please offer the Ethic Committee Approval in the "Material and Methods" chapter.
Regarding the statistical methods you apply, were all the data with normal distribution to use only Student's test? If yes, you must specify this.
Response:
The F distribution confirmed in an F test that compares the degree of variability in two groups. The specify added in the text (lines 185-186 in red).
Conclusions
Please write the conclusions with more details. Based on your results you need to mention the molecules influenced by pteryxin action and how. Anyhow the first sentence of your conclusions (line 307-308) is confusing, please reformulate it.
Response:
As you suggested, the conclusion section rewrote in detail about the action of pteryxin obtained from the results.
Round 2
Reviewer 1 Report
The manuscript is much clear now, and it reads quite well. It should be published and will certainly be welcomed by the scientific community.
Reviewer 2 Report
The authors made the corrections I recomended.